# Retrieving Tarnished Daguerreotype Content Using X-ray Fluorescence Imaging—Recent Observations on the Effect of Chemical and Electrochemical Cleaning Methods

**Alison Stark** [1], **Fraser Filice** [1], **James J. Noël** [1], **Ronald R. Martin** [1], **Tsun-Kong Sham** [1,*], **Yanhui Zou Finfrock** [2,3] **and Steve M. Heald** [3]

1    Department of Chemistry, Western University, London, ON N6A 5B7, Canada; astark9@uwo.ca (A.S.); ffilice@uwo.ca (F.F.); jjnoel@uwo.ca (J.J.N.); rrhm@uwo.ca (R.R.M.)
2    CLS@APS, Science Division, Canadian Light Source, Saskatoon, SK S7N 2V3, Canada; zou.finfrock@lightsource.ca
3    Advance Photon Source, Argonne, National Laboratory, Argonne, Lemont, IL 60439, USA; heald@anl.gov
*    Correspondence: tsham@uwo.ca

**Abstract:** We report a study on the effect of chemical and electrochemical cleaning of tarnished daguerreotypes observed using X-ray fluorescence (XRF) microscopy with a micro-focussed X-ray beam from a synchrotron source. It has been found that, while both techniques result in some success depending on the condition of the plate and the experimental parameters (chemical concentration, voltage, current, etc.) the effect varies, and cleaning is often incomplete. The XRF images using Hg $L_{\alpha,\beta}$ at an excitation energy just above the $L_3$ edge threshold produce fine images, regardless of the treatment. This finding confirms previous observations that if the bulk of the image particles remains intact, the surface tarnish has little effect on the quality of the original daguerreotype image retrievable from XRF.

**Keywords:** daguerreotype; tarnish; chemical cleaning; electrocleaning; synchrotron; X-ray fluorescence imaging

## 1. Introduction

### 1.1. The Daguerreotype

Daguerreotypes are the earliest form of photography, produced on a silver-coated copper plate, which was used predominantly in the mid to late 1800s [1,2]. The process of creating a daguerreotype was developed by Louis-Jacques-Mande Daguerre and perfected in 1839. As creating daguerreotypes was a lengthy and costly practice, only people of high status could afford to have their photo taken this way [3,4]. Nevertheless, daguerreotypes offer snapshots, literally, for the very first time, into this era of human activities, and are of artistic, cultural, and historical significance. Efforts to preserve and restore these images have aroused considerable interest in the art preservation community [5].

The production of a daguerreotype image requires several steps. The result is a high contrast, one-of-a-kind photograph [1–4]. The process begins with making a finely polished silver-coated copper plate. This is followed by the exposure of the surface to iodine, making the plate photosensitive upon the formation of silver iodide. Later variations utilized alternative halogens, chlorine, bromine, or a combination of these, in order to increase the sensitivity of the surface to light. The photosensitive plate is mounted in the lightproof interior of a camera. When the photo to be taken is appropriately framed, the lens cap is removed, exposing the plate to light. This step causes the formation of silver image particles that result from the photolysis of the silver halide on the silver surface, creating the image. Areas with dense distributions of image particles of a rather consistent shape and size correspond to a high light intensity, whereas areas exposed to a low light intensity

display thin and nonuniform image particles. Bright regions are the result of the particles scattering light, and where there are few image particles, only light from specular reflection can be seen. This is why the light and dark areas can change if the daguerreotype is tilted [1]. After the image has formed on the surface, the plate is then exposed to mercury vapor, which fixes the image on the silver-coated copper plate. This is the crucial step in the entire process and, as we shall show below, the presence of Hg on the image's particles allows us to retrieve the fine image details from a tarnished daguerreotype. Excess halides are removed with a salt solution, such as sodium thiosulfate. This makes the surface insensitive to light, and halts the creation of additional image particles, which could cloud the image [1,2]. Next, the silver-coated copper plate is washed with distilled water, and a gold chloride solution is poured on the daguerreotype to ensure the longevity and durability of the image [1]. Note, the addition of gold chloride, known as the gilding step, was later introduced into the final stages of the daguerreotype procedure. Finally, the plate is heated to dry the surface. This process produces a completed daguerreotype image of the subject.

### 1.2. Deterioration of the Daguerreotype

Daguerreotypes are subject to the formation of surface tarnish, which, in the extreme, can completely obscure the image. Surface corrosion will alter the shape and refractivity of the image particles, and hence the direction and intensity of the scattered light. The most frequent tarnishes are silver halides, silver oxides, and sulfur compounds originating from incomplete washing during the original preparation, exposure to atmospheric gases, and deterioration of the cover glass [5,6]. Studies have also focused in great detail on the effects of the daguerreotype storage and exposure conditions. Extreme temperatures and/or humidity can have a negative influence on the integrity of the surface. When a daguerreotype is stored, a glass cover is usually present on the image side of the surface; however, deterioration of the glass cover can be another factor contributing to the degradation of the plate [7]. Many original glass cover plates contained sodium and potassium, which have been shown, over time, to diffuse from the glass and leave deposits on the daguerreotype surface. This leads to highly localized spots of corrosion across the daguerreotype. In addition, the glue that was used to adhere the glass cover to the plate also contributed to corrosion at the perimeter of the daguerreotypes [8,9]. Compounds such as oxides and various sulfides may have been formed because of this practice [10].

### 1.3. Conservation and Preservation Methods

Daguerreotypes are fragile, and Daguerre himself recommended that the plate be protected with a cover glass. About 20 years after their invention, the commercial production of daguerreotypes ceased; there was little effort to preserve them and they became collectors' items. It was not until the twentieth century that archives and art institutions began to collect and preserve daguerreotypes [1]. Many preservation and restoration processes have been tried with varying success. As a result of the variation in the methods of preparation, as well as diverse storage conditions, many daguerreotypes have unique damage requiring tailored cleaning methods. As a result, there is not a single method guaranteed to restore these images. While progress has been made towards a more universal cleaning procedure, these procedures completely depend on the original quality of the surface [5]. There are two general cleaning techniques: chemical cleaning and electrocleaning [1,2]. There are also two common methods in electrocleaning, sometimes referred to as the Wei method and the Barger method [11,12]. The Wei method simply applies a cathodic polarization to the daguerreotype plate in a cleaning solution [11]. The Barger method applies both oxidizing and reducing polarizations to induce anodic and cathodic currents on the daguerreotype, switching between the two throughout the process. It is hoped that by manipulating the surface chemistry, the tarnish will be removed, while the image particles (nano particles of Ag coated with Hg forming an amalgam) will remain intact. Various studies have proven that both methods can help restore the daguerreotype image to some extent. However, as

each daguerreotype is unique in terms of the elemental composition of the tarnish and how deteriorated it is, the methods are not always effective. In some cases, electrocleaning treatments have further damaged the daguerreotype. To remove the tarnish from the surface with electrocleaning methods, adjustments of the potential are made. This potential is biased on the daguerreotype surface, and monitored by a reference electrode, as described below.

### 1.4. XRF Imaging Using a Micro Focused X-ray Beam

It was recently reported that using a micro-focused X-ray beam from a synchrotron light source and selecting the excitation energy to just above the $L_3$ absorption edge of Hg and tracking the intensity of the Hg $L_\alpha$ and $L_\beta$ lines in a two-dimensional scan across the daguerreotype plate could retrieve the original image from the daguerreotypes tarnished beyond recognition [7]. These results show that it is the integrity of the silver image particles that were formed upon the photochemical reaction of the photosensitizer, silver halide, when exposed to the object, and the subsequent exposure to hot Hg vapor that determines the quality of the daguerreotype [1,2]. Thus, Hg vapor stabilizes the image particles of silver clusters, forming an amalgam that defines the image, and the image particles are preserved by the presence of Hg. If surface reactions or adventitious contaminants such as the organic molecules that tarnish the plate only affect the surface and the near-surface region (on the order of < nm), this would have very little effect on the image retrieved from Hg $L_\alpha$ fluorescence X-rays. This is also why chemical and electrocleaning would work if the bulk of the image particles remained largely undisturbed after cleaning. The objective of this work is to proceed with chemical cleaning and electrocleaning methods under various conditions on a single daguerreotype plate, and to then conduct Hg XRF imaging to further confirm this notion [3,7].

## 2. Materials and Methods

### 2.1. The Daguerreotype Plate and Cleaning Solutions

The daguerreotype plate under investigation, "Little Girl, Pretty Purse", was provided by the Canadian Conservation Institute, National Gallery of Canada (Ottawa, Canada). The reagents used for the chemical cleaning portion included reagent grade ammonium hydroxide ($NH_4OH$, Caledon Laboratory Chemicals) and reagent grade sodium thiosulfate $Na_2S_2O_3$, 99% assay (Sigma Aldrich). Solutions containing specific reagent concentrations were prepared, including a sodium thiosulfate solution of 3% mass/volume (0.190 M) and 1% and 0.5% mass/volume of ammonium hydroxide (0.294 M and 0.147 M, respectively). Additionally, 0.01 M and 0.1 M reagent grade potassium chloride (KCl BioShop), and 0.01 M reagent grade potassium sulfate ($K_2SO_4$, 99% assay, Caledon Laboratory Chemicals) solutions were created for the electrochemical cleaning process. All of the dilution schemes and rinsing processes were done with Type 1 water (18.2 M$\Omega \times$ cm resistivity).

### 2.2. Chemical and Electrochemical Solution Cell Fabrication

To clean the daguerreotype, two cells were designed using Autodesk Inventor Professional 2020, and were 3D printed from UV-cured resin (ANYCUBIC Photon UV 3D Printer, ANYCUBIC 405 nm Resin Green) for chemical and electrochemical cleaning. These cells were designed to be clamped against the face of the daguerreotype, compressing an O-Ring and creating a tight seal with the surface. Both designs exposed a small surface area of the sample surface to the solution, allowing for multiple reagents and techniques to be tested on localized regions of the same specimen. The chemical cleaning device was composed of three solution wells in proximity, allowing for the simultaneous testing of multiple solutions on a localized area. The electrochemical cell incorporated a much larger solution volume, accommodating a three-electrode electrochemical setup [13]. These arrangements are displayed in Figure S1.

### 2.3. Chemical Cleaning and Electrocleaning

The chemical cleaning cell was clamped onto the daguerreotype surface, as shown in Figure S1c. Each of the three wells had a different solution pipetted into them and were left for 1 h. After that, the solutions were removed, and each well was gently rinsed with Type 1 water. The daguerreotype was then rinsed and patted dry with Kimwipes. Each site was then inspected at Surface Science Western with a VHX-6000 optical microscope (Keyence), as well as VP-SEM (Hitachi SU3900) and EDX (Oxford Instruments Ultim Max 65).

Electrocleaning and measurements were performed with a Solartron 1287 potentiostat on the daguerreotype plate. A three-electrode setup was used, as shown in Figure S1a,b, where the daguerreotype was the working electrode, Ag/AgCl was the reference electrode, and platinum foil was the counter electrode. The solution cell was clamped onto the daguerreotype, then filled with one of the several electrolytes being studied, and a 5 min open circuit potential (OCP) measurement was performed. The OCP measurement determines the resting potential of the working electrode. This was followed by an electrolytic cleaning step using the Wei method [11] or the Barger method [12], described as follows.

In the Wei method, we applied a constant cathodic polarization (constant negative potential) for 90 s, as seen in the potential versus time graph (Figure 1, top-left). In the corresponding current versus time graph (Figure 1, top-right), a typical current response is shown; the current started at a rather negative value, while reducible species were abundant on the surface, and then gradually approached zero, suggesting that the oxidized surface species were becoming depleted as the cleaning procedure progressed.

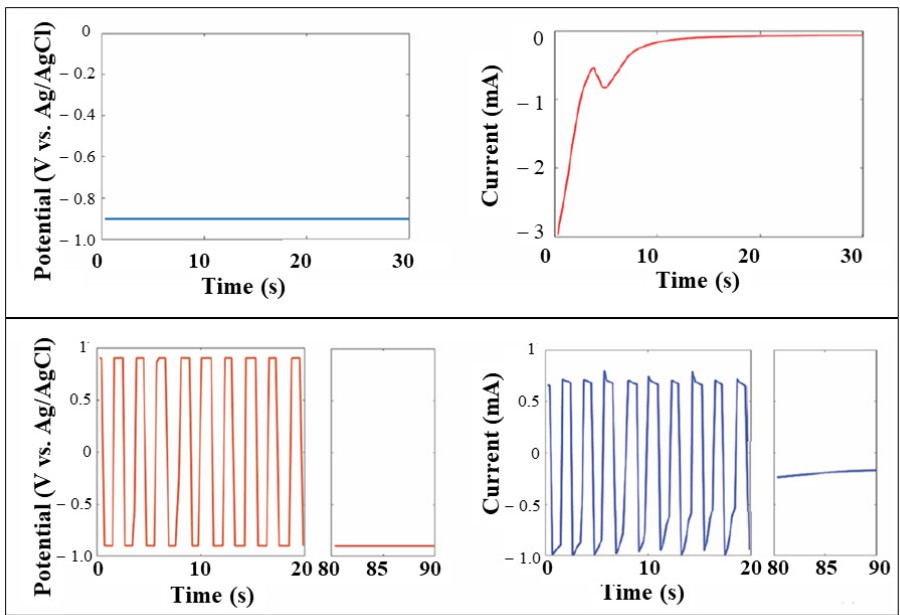

**Figure 1.** (**Top**): the potential versus time, and current versus time profiles for the Wei method; only the first 30 s were plotted in a 90-s test. (**Bottom**): the potential versus time, and current versus time profiles for the Barger method; only the first 20 s were plotted in an 80-s test, including the 10 s of the constant negative potential applied at the end of each test.

In the Barger method, we applied a modified version in which the applied potential was alternated between anodic and cathodic polarizations in 2-s intervals for 80 s, followed by a 10-s cathodic cleaning step, also seen in Figure 1, bottom-left, in the potential versus time graph. The corresponding current versus time graph (Figure 1, bottom-right) does not show the same approach to zero current seen in the Wei method, for several reasons. First, the electrode was not given enough time during any of the cathodic stages to achieve a steady state. Second, the anodic phase preceding each cathodic phase of the oscillation generated more oxidized species [7] for reduction during the subsequent cathodic phase,

and finally, the current during the anodic phase should never be eliminated because it could correspond to the oxidation of the Ag that makes up the bulk of the daguerreotype. Following the application of one of these cleaning profiles, the electrolyte solution was immediately removed, and the daguerreotype surface was gently flushed with Type 1 water. Each site was then analyzed optically, as well as by VP-SEM and EDX.

## 2.4. SEM and EDX Characterization

The morphology and elemental distribution of the plate before and after the cleaning were examined with SEM and EDX, respectively [14]. A Hitachi SU3900 Large Chamber Variable Pressure SEM combined with an Oxford ULTIM MAX 65 SDD X-ray analyzer was used. High resolution (up to 100 k X magnification) [15] FE-SEM imaging was performed using a Hitachi SU8230 Regulus Ultra High-Resolution Field Emission SEM. Selected areas on the daguerreotype were imaged using FESEM (image resolution of 0.6 nm at 15 kV acceleration or 0.8 nm at 1 kV acceleration).

## 2.5. XRF Imaging Using Synchrotron Radiation

XRF images of the plate were recorded at the microprobe station at CLS@APS at the ID beamline of sector 20 of the Advanced Photon Source at Argonne National Laboratory [16]. The ID line was equipped with a Si(111) double crystal monochromator and a KB mirror capable of focussing the X-ray beam down to 5 micrometres routinely. We used an excitation energy of 13 keV; this energy is just above the Hg $L_3$ (12,284 eV) and Au $L_3$ edge (11,919 eV), producing Hg $L\alpha_{1,2}$ (9989 eV and 9898 eV) and Au $L\alpha_{1,2}$ (9713 eV and 9628 eV) X-ray fluorescence lines, as well as other lines of interest, e.g., $K_\alpha$ of first row transition elements [17]. The incident beam was tuned to a spot size of ~ 50 μm to optimize the data acquisition efficiency. The experiment was conducted when APS was running in a top-up mode, 24 bunches with a total current of 100 mA. This mode, together with the incident focussed beam ($I_o$) being monitored with an ionization chamber, ensured the beam stability and proper normalization, which is essential as it normally takes several hours to scan the entire plate. In this run, the photon flux was approximately $10^{12}$ photons per second over a spot size of ~50 μm with a step size of 50 μm. The illumination time was 50 ms per pixel, number of pixels of the map was 881 X 1061, and the total scan time was 13 h 24 min and 23 s. It should be noted that synchrotron XRF imaging has been widely used and the scope of its application can be found in a recent contribution [18].

The daguerreotype plate was mounted on a three-axis platform and the scanning was done by moving the plate across the beam pixel by pixel. The XRF image was obtained by setting the energy window of interest, e.g., Hg $L_\alpha$ and Au $L_\alpha$, in the X-ray fluorescence spectra collected by a four-element Vortex-Me4 solid state detector (~250 eV energy resolution) and stored in a multichannel analyser (MCA). The experimental set-up is shown in Figure S2, and a snapshot of the MCA display is shown in Figure S3, where the fitting of the overlapping fluorescence X-rays with which we could obtain more accurate elemental distribution and better contrast were obtained.

## 3. Results

The optical images of the "Little Girl, Pretty Purse" daguerreotype before cleaning and after cleaning, as well as the XRF image from collecting Hg $L_\alpha$ fluorescence X-rays, are shown in Figure 2. It is apparent from Figure 2 that the chemical and electrochemical cleaning methods are generally effective. We will inspect several selected areas and discuss the effects below.

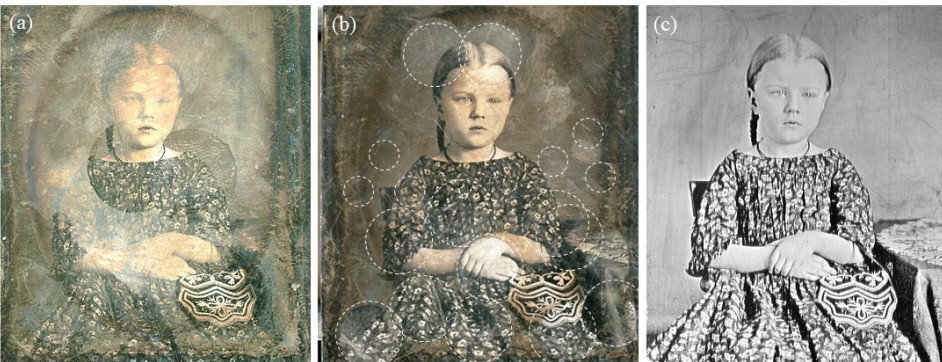

**Figure 2.** (**a**) Optical image of the plate before cleaning, the oval marks are due to some preliminary cleaning test using similar solution cell prior to this experiment. (**b**) Optical image after a series of local cleaning attempts with chemical (two regions marked by three wells of the solution cell) and electrochemical methods using larger oval shape cells (see text). (**c**) XRF image obtained with Hg $L_\alpha$ fluorescence X-rays.

### 3.1. Chemical Cleaning

Let us examine the three different sites cleaned using the three-well cell assembly (small oval marks) on the left of Figure 2b, each with its own solution, namely: 3% sodium thiosulfate, 1% ammonium hydroxide, and 0.5% ammonium hydroxide. An ammonium hydroxide solution was chosen as it removes halides from the surface by forming soluble ammonia silver complexes, and sodium thiosulfate was historically used in the production of daguerreotypes to remove silver-halides from the material surface. As the haze on a daguerreotype surface is normally attributed to halide formation, sodium thiosulfate would also be a good chemical cleaning method in addition to ammonium hydroxide [2]. The results of the chemical cleaning are closely examined in Figure 3.

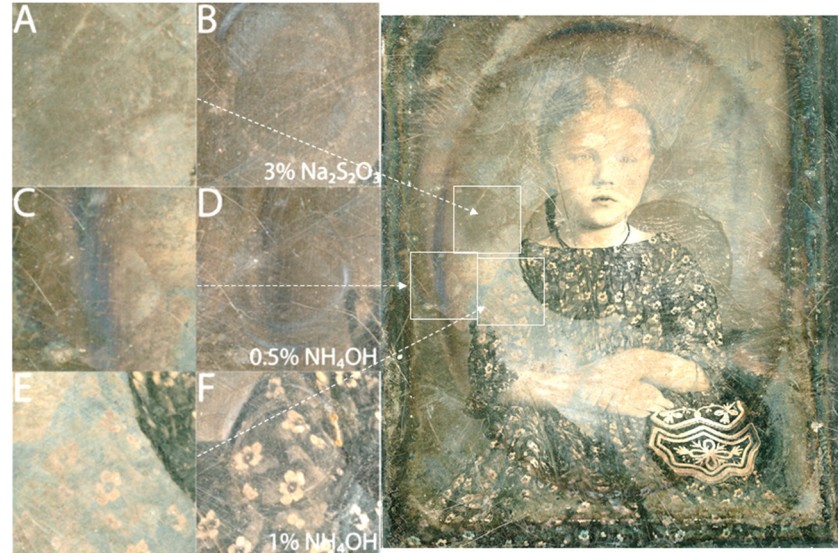

**Figure 3.** Optical images of the plate before (**A,C,E**) and after local cleaning (**B,D,F**) with solutions, as marked. The locations were situated on the optical image of the plate before cleaning.

From the left panel of Figure 3, we see that all cleaning solutions were successful in removing the foggy haze from the surface. The 3% $Na_2S_2O_3$ solution produced a noticeable reduction in surface clouding (Figure 3A,B). A similar increase in sample clarity was observed with the application of 0.5% $NH_4OH$ (Figure 3C,D). The 1% $NH_4OH$ (Figure 3F) uncovered the masked floral print with fine details and good contrast. EDX maps (Figure S4) revealed

the presence of Ag, Au, Hg, S, and Cl, as well as Hg-coated image particles that were slightly less than a micrometre. After cleaning, they remained intact, while the Cl and S signals were reduced.

### 3.2. Electrochemical Cleaning with Cathodic Method

A preliminary testing with silver-coated copper wires was performed to ensure that the applied potentials would not damage the daguerreotype. Although previous studies employed much higher potentials in their electrochemical cleaning procedures, our initial testing indicated that a potential of −0.9 V would be sufficient to cathodically clean the surface (the Wei method) [11] without causing any noticeable surface damage.

To test the effectiveness of the cathodic electrochemical cleaning methods, we used different electrolytes that were applied to selected sites, as illustrated in Figure 4, where the optical images before and after cleaning are shown. The Wei method involved the application of a constant negative potential (Figure 1) to the daguerreotype plate, therefore constantly reducing the surface. The electrolytes were selected such that they had a negligible chemical cleaning effect. Then, 0.01 M, 0.1 M KCl, and 0.01 M $K_2SO_4$ solutions were used. All of the electrochemically cleaned sites showed an optically improved image (Figure 4B,D,F, left panel). Most remarkably, the hands (Figure 4E,F, left panel) revealed greater details than before any cleaning attempts.

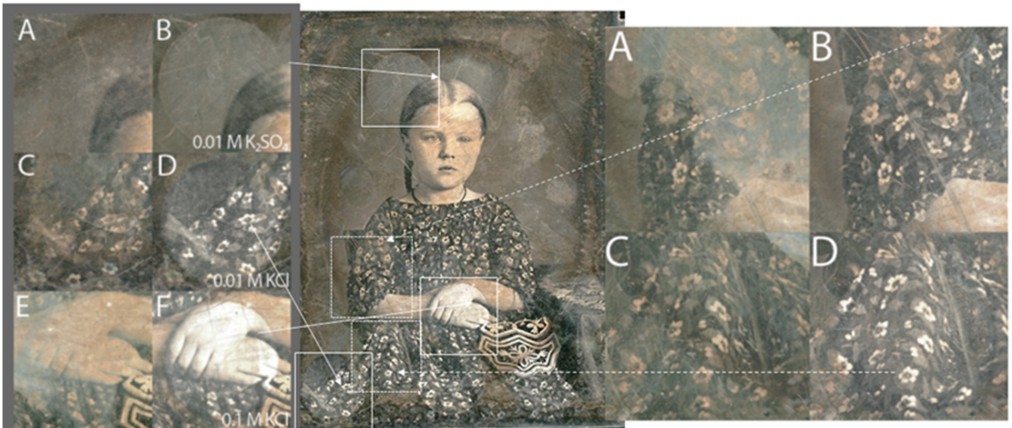

**Figure 4.** Left panel: Optical images (**A**,**C**,**E**) before and (**B**,**D**,**F**) after electrochemical cleaning using the Wei method with an electrolyte solution: B, 0.1 M KCl; D, 0.01 M KCl; and F, 0.01 M $K_2SO_4$. Mid panel: Optical image after all of the cleaning processes. The regions of interest are marked with squares. Right panel: Optical images before (**A**,**C**) and after (**B**,**D**) electrochemical cleaning with 0.01 M KCl as the electrolyte. Image B, Barger method (−0.9 V to 0.9 V), and image D, Barger method (−1.2 V to 1.2 V).

It should be noted, however, that cleaning with a KCl solution (0.1 M), introduced Cl⁻ ions into the solution. This site showed a greater amount of residual white haze (formation of AgCl) following cleaning than the site using the lower concentration KCl solution (0.01 M, Figure 4D, left panel). This was the result of the common ion effect reducing the solubility of AgCl. An improved image is observed for Figure 4B, left panel, where 0.01 M $K_2SO_4$ solution was used. All three sites showed great improvement optically after only 90 s of cleaning. EDX maps were also obtained before and after cleaning, showing results like those obtained from the chemical cleaning experiment noted above.

### 3.3. Electrochemical Cleaning Using Chemical Cleaning Solutions

We have also explored the effect of using chemical cleaning solutions, such as $NH_4OH$ and $Na_2S_2O_3$, as the electrolytes for the electrochemical cleaning procedures with both the Wei and Barger methods. In Figure 5, the optical images of the daguerreotype before and after the application of the combined electrochemical and chemical cleaning are

shown. The improvement of the overall image quality when cleaning was performed with a solution of 0.15% $NH_4OH$ can clearly be seen. This procedure was applied with a 0.19% $Na_2S_2O_3$ solution with similar results. Figure 5B,D, right panel, correspond to the Barger method with an applied potential range of $-0.9$ V to 0.9 V, and $-1.2$ V to 1.2 V versus Ag/AgCl, respectively. Again, all of the images after cleaning showed more detail than before.

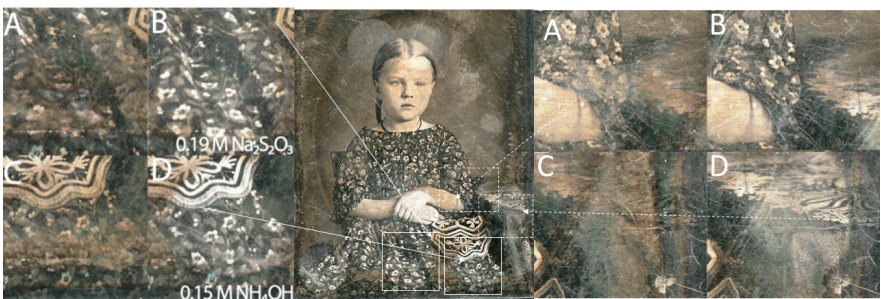

**Figure 5.** Left panel: Optical images (**A**,**C**) before and (**B**,**D**) after electrochemical cleaning using the Wei method with an electrolyte solution: B, 0.19 M $Na_2S_2O_3$; D, 0.15 M $NH_4OH$. Middle panel: Optical image after all of the cleaning processes. The regions of interest are marked with squares. Right panel: Optical images before (**A**,**C**) and after (**B**,**D**) electrochemical cleaning with 0.5 M $NH_4OH$. Image B, Barger method ($-0.9$ V to 0.9 V), and image D, Barger method ($-1.2$ V to 1.2 V).

For example, in Figure 5C, right panel, it is difficult to see what is present, but after cleaning it becomes apparent that there is a very intricate table cloth (Figure 5D, right), where many details within it are now evident. According to the EDX maps, both the Barger and Wei method can remove both the halides and sulfides present on the surface.

### 3.4. XRF Imaging

A closer look at Figure 2 shows the optical images before and after cleaning, as well as the Hg $L_\alpha$ image. It is apparent from Figure 2C that the XRF image has revealed a finely detailed portrait of the little girl and the pretty purse with a very clean background, as if all the tarnish were removed. The most noticeable difference is that, while the various cleaning methods applied to the plate show nonuniform cleaning in the optical image depending on the cleaning condition and the region of interest, removal of tarnish from the daguerreotype is complete in the XRF image, which reveals fine details everywhere across the portrait.

The XRF image can be fitted using the software package PyMCA [19,20], in which the X-ray fluorescence peaks are fitted and the area under the curve is the intensity contributing to the image. This procedure removes contributions from the overlapping peaks, such as Au $L_\alpha$ in the case of Hg $L_\alpha$. An XRF image can also be obtained using Hg $L_\beta$, Au $L_\alpha$, and Au $L_\beta$. The images from the Hg L PyMCA fit, and Hg $L_\beta$ and Au $L_\alpha$ without fitting, are shown in Figure 6. It is interesting to note that the image is also finely revealed in the Hg $L_\beta$ map and is noticeable in the Au map, albeit thinly veiled. The presence of a veiled Au image indicates that a gilding step using a gold chloride solution was applied after the image particles were formed and fixed when the plate was made, so that Au was found all over the plate, tracking how the gliding was done at the time. An image can still be observed from the Au fluorescence, suggesting that Au tracks the density and the distribution of the image particles, as well as the featureless region of the Ag plate.

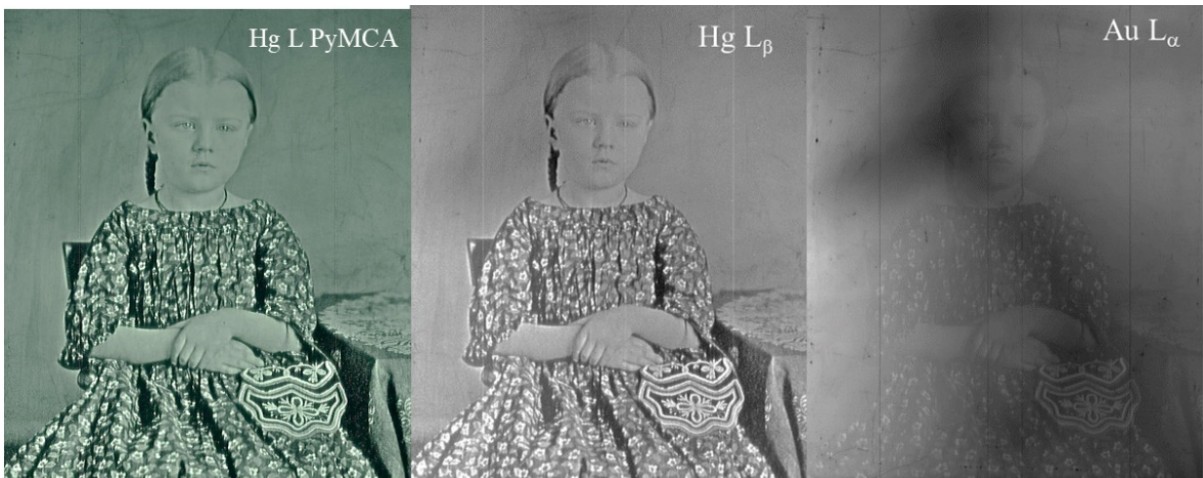

**Figure 6.** XRF image from left to right using the Hg L PyMCA fit, and Hg L$_\beta$ and Au L$_\alpha$ without fitting.

When comparing the Hg L$_\alpha$ image in Figure 2C with the Hg L PyMCA and L$_\beta$ images (Figure 6), it appears that while the L$_\alpha$ and L$_\beta$ images without fitting were of equally good quality, the PyMCA image showed a slightly better spatial resolution and contrast at closer scrutiny. This is because the Hg and Au L$_\alpha$ lines could not be completely resolved with the solid-state detector (SSD) without fitting. We also tracked the Cu K$_\alpha$ line, which did not show any image as the signal came from the Cu plate, and Cu was not involved in the formation of image particles. The Ag L emission was too weak at this excitation energy to be detected, and did not reveal any image either. It will be of interest to track Ag with tender X-ray excitation at just above the Ag L$_3$ edge (3351 eV) [3].

## 4. Conclusions

We conducted chemical and electrochemical cleaning procedures on various sites from a single, partially tarnished daguerreotype. We found that both methods of cleaning were effective at removing the tarnish and restoring the image. While all procedures seemed to improve the image by some degree, the tarnish was never removed uniformly or entirely. The chemical cleaning procedures were sufficient to remove the halides/white haze from the surface. Optical images taken after cleaning still appeared to have areas with a brown/orange tinge on the surface. Electrochemical cleaning was sufficient at removing the sulfides from the surface in addition to the halides, and did it faster. Both the Barger and the Wei electrocleaning methods improved the visual appearance of the image. Again, they did not always remove the orange/brown tarnish colour form the surface. As noted in the introduction, daguerreotypes can vary significantly depending on when they were made, the methods and equipment of the artist who made them, and the conditions under which they were stored. By performing these experiments on small regions of the same daguerreotype, we tried to obtain the most consistent possible initial conditions for comparison. The effectiveness of the treatment will depend entirely on the original integrity of the daguerreotype. They are not uniform. This work confirms previous observations that the only sure way to retrieve the complete contents of the daguerreotype is through synchrotron radiation X-ray fluorescence imaging. This technique will ensure that events of historical significance from a tarnished plate can be retrieved. With the XRF method, even if the daguerreotype is severely tarnished, provided there is still sufficient mercury on the image particles on the surface, the image in its entirety can still be reconstructed through digitizing the XRF images. The cleaning methods have been shown to improve the image optically; nevertheless, this should be undertaken with caution. Once the Hg is gone, the image will be lost forever. To refine the daguerreotype cleaning methods even more, further research would need to be performed to determine the detailed chemical

composition of the substrate plate, the image particles, the tarnish, and its interplay with the environment, such as the daguerreotype housing and the protecting glass.

**Supplementary Materials:** The following are available online at https://www.mdpi.com/article/10.3390/heritage4030089/s1. Figure S1: Experimental setup for electrochemical cleaning and chemical cleaning. (a) Schematic for the three-electrode set up for electrocleaning. (b) Actual set up for electrocleaning; the area of interest is confined by the perimeter of the cell, which leaves an oval mark on the plate after cleaning. The working electrode (daguerreotype), counter electrode (Pt), and reference electrode (Ag/AgCl) are noted. (c) Setup for chemical cleaning with the three-well cell clamped down on the plate. This setup leaves behind three small oval marks on the plate. (see Figure 2, middle panel and text). Figure S2: Experimental arrangement for the XRF imaging. The focussed beam (yellow line from left to right) with a spot size of 30 μm × 20 μm is stationary. The plate is mounted on a three-axis stage that moves the plate across the beam with submicrometre precision, pixel by pixel. The fluorescence X-rays are collected with a four element SSD (VotexME4). The data are stored in a multichannel analyzer (MCA). Desired energy windows are set to collect element-sensitive maps (see Figure S4). Figure S3: A snapshot of the MCA display during a scan (top); the abscissa is photon energy and the ordinate axis is intensity in a semi-log plot. The Hg L intensities fit using PyMCA are shown in the bottom panel (both $L_\alpha$ and $L_\beta$ are used, black dotted curve). Figure S4: EDX maps of Ag, S, Au, Hg, and Cl, and the backscattered (BSE) SEM image (black and white) for the chemical cleaning solutions discussed in Figure 3A,B. A: before cleaning; B: after 3% $Na_2SO_3$.

**Author Contributions:** Conceptualization, R.R.M., J.J.N., and T.-K.S.; methodology, A.S., F.F., J.J.N., Y.Z.F., and T.-K.S.; lab experiment and analysis, A.S. and F.F.; A.S. is an undergraduate student under the supervision of J.J.N., R.R.M., and T.-K.S.; synchrotron, XRF imaging experiment and analysis, Y.Z.F., S.M.H., and T.-K.S.; writing—original draft, T.-K.S.; writing—review and editing, all of the authors. All of the authors have read and agreed to the published version of the manuscript. All authors have read and agreed to the published version of the manuscript.

**Funding:** This research was supported by the Natural Science and Engineering Research Council of Canada (NSERC, RGPIN-2019-05926), the Canada Research Chair Program (CRC), the Canada Foundation for Innovation (CFI), and the University of Western Ontario (UWO). This research used resources of the Advanced Photon Source, an Office of Science User Facility operated for the U.S. Department of Energy (DOE) Office of Science by Argonne National Laboratory, and was supported by the U.S. DOE under contract no. DE-AC02-06CH11357.

**Institutional Review Board Statement:** Not applicable.

**Informed Consent Statement:** Not applicable.

**Data Availability Statement:** Data used in this work are available from the authors upon request.

**Acknowledgments:** CLS@APS where XRF imaging was conducted is an operation of the Canadian Light Source, which is supported by the Canada Foundation for Innovation (CFI), the Natural Sciences and Engineering Research Council (NSERC), the National Research Council (NRC), the Canadian Institutes of Health Research (CIHR), the Government of Saskatchewan, and the University of Saskatchewan.

**Conflicts of Interest:** The authors declare no conflict of interest.

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
