# Peer review of "Retrieving Tarnished Daguerreotype Content Using X-ray Fluorescence Imaging—Recent Observations on the Effect of Chemical and Electrochemical Cleaning Methods"

_heritage, doi:10.3390/heritage4030089_

Round 1
Reviewer 1 Report
The manuscript is clear and well written. The findings are useful and informative to conservation scientists and to the cultural heritage and the synchrotron imaging communities. It would be helpful if the authors could provide the XRF scan parameters, i.e., photon flux on the beam spot, illumination time per pixel, number of pixels for the image and total scanning time (including illumination time and potential dead times from moving, readout, etc). As XRF elemental mapping with synchrotron radiation and lab-based instruments has been carried out extensively in the cultural heritage community and is a growing field, the authors should consider quoting some of the original and current work, as well as reviews, other than their own papers.
Author Response
The scanning parameters requested by the reviewer in terms of flux, spot size, time etc. are now included in the text in section 2.5 (XRF imaging...) on page 5; a new reference is also added to show the scope of synchrotron XRF imaging in art and archaeology (ref.18)
Reviewer 2 Report
The manuscript reports on the study of the effect of chemical, electrochemical and X-ray fluorescence on the capacity to clean and recover Daguerreotype photos produced in the nineteenth century. Chemical cleaning was performed with different solutions applied to several parts of the photo. The same was done with the electrochemical cleaning, where the Wei and Barger methods were used. A before-after comparison was made in each part of the treated photo to access the efficiency of each method. Additionally, X-ray fluorescence was measured from different parts of the photo, detecting Hg and Au lines. The XRF image was treated with the PyMCA software. It was observed that the XRF signal from Hg reproduced the image with good resolution even from less cleaner parts. The manuscript has a detailed work, original results and only needs minor revisions. I have the following comments:
- Several times along the text the authors refer to EDX results but they are not shown in the manuscript. For example, in page 6 it is referred that “EDX maps (not shown) reveal the presence of Ag, Au, Hg, S and Cl, as well as Hg-coated image particles which are slightly less than a micrometre. After cleaning, they remain intact, while Cl and S signals are reduced.”. Information like this is important for the results and it would be interesting to for the reader that they are presented, to follow the discussion, even if they are only put in the supplemental information.
- While comparing figures 2c and 6, with the XRF results, it is written on page 9 that “It appears that while the Lalpha and Lbeta images without fitting are of equally good quality, the PyMCA image shows a slightly better spatial resolution and contrast at closer scrutiny.”. On the other hand, in the legend of figure 2 it is referred that “vertical white lines are artifacts due to a glitch in data collection”. Could this be also influencing the perceived spatial resolution referred above ?
Author Response
The reviewer asked to see some EDX maps; in response, we have included a typical set of EDX for a selected region before and after cleaning in the supplementary materials as Figure S4.
The reviewer also question the vertical whitelines we noted in the legend of Figure 2; there whitelines appeared during data acquisition and are hardly in Figure 2(c) and Figure 6. We left it out of the legend in the revision. They have no bearing on the comparison between Figure 2(c) and Figure 6.
Reviewer 3 Report
The topic of the reviewed paper is extremely relevant, which is associated with significant negative changes occurring with daguerreotypes during their long-term storage in non-optimal conditions. Of undoubted interest is the assessment of the state of daguerreotypes from museum and private collections, as well as family archives. The reviewed paper is devoted to the analysis and improvement of technologies for the restoration of tarnished daguerreotypes. To preserve existing daguerreotypes as objects of cultural heritage and increase their accessibility to a wide range of researchers and people interested in history, it is necessary not only to improve technologies for the restoration of daguerreotypes, but also to develop a databases of digital optical images and XRF-images of daguerreotypes.
Author Response
We thank the reviewer for the positive comments. No revision was suggested by this reviwer